# Assessment of Fatty Acid and Oxylipin Profile of Resprouting Olive Trees Positive to *Xylella fastidiosa* subsp. *pauca* in Salento (Apulia, Italy)

**DOI:** 10.3390/plants13162186

**Published:** 2024-08-07

**Authors:** Valeria Scala, Marco Scortichini, Federico Marini, Dario La Montagna, Marzia Beccaccioli, Kristina Micalizzi, Andrea Cacciotti, Nicoletta Pucci, Giuseppe Tatulli, Riccardo Fiorani, Stefania Loreti, Massimo Reverberi

**Affiliations:** 1Council for Agricultural Research and Economics (CREA), Research Centre for Plant Protection and Certification, 00156 Roma, Italy; nicoletta.pucci@crea.gov.it (N.P.); giuseppe.tatulli@crea.gov.it (G.T.); riccardo.fiorani@crea.gov.it (R.F.); stefania.loreti@crea.gov.it (S.L.); 2Council for Agricultural Research and Economics (CREA), Research Centre for Olive, Fruit and Citrus Crops Research Centre for Olive, Fruit and Citrus Crops, 00134 Roma, Italy; marco.scortichini@crea.gov.it; 3Department of Chemistry, Sapienza University of Rome, 00185 Roma, Italy; federico.marini@uniroma1.it; 4Department of Environmental Biology, Sapienza University of Rome, 00185 Roma, Italy; dario.lamontagna@uniroma1.it (D.L.M.); marzia.beccaccioli@uniroma1.it (M.B.); kristina.micalizzi@uniroma1.it (K.M.); andrea.cacciotti@uniroma1.it (A.C.); massimo.reverberi@uniroma1.it (M.R.)

**Keywords:** OQDS, crown restoration, satellite imagery, fatty acid oxylipins, defense-related hormones, NDVI

## Abstract

*Xylella fastidiosa* subsp. *pauca* ST53 (XFP), the causal agent of olive quick decline syndrome (OQDS), was thoroughly investigated after a 2013 outbreak in the Salento region of Southern Italy. Some trees from Ogliarola Salentina and Cellina di Nardò, susceptible cultivars in the Gallipoli area, the first XFP infection hotspot in Italy, have resprouted crowns and are starting to flower and yield fruits. Satellite imagery and Normalized Difference Vegetation Index analyses revealed a significant improvement in vegetation health and productivity from 2018 to 2022 of these trees. Lipid molecules have long been recognized as plant defense modulators, and recently, we investigated their role in XFP-positive hosts and in XFP-resistant as well as in XFP-susceptible cultivars of olive trees. Here, we present a case study regarding 36 olive trees (12 XFP-positive resprouting, 12 XFP-positive OQDS-symptomatic, and 12 XFP-negative trees) harvested in 2022 within the area where XFP struck first, killing millions of trees in a decade. These trees were analyzed for some free fatty acid, oxylipin, and plant hormones, in particular jasmonic and salicylic acid, by targeted LC-MS/MS. Multivariate analysis revealed that lipid markers of resistance (e.g., 13-HpOTrE), along with jasmonic and salicylic acid, were accumulated differently in the XFP-positive resprouting trees from both cultivars with respect to XFP-positive OQDS symptomatic and XFP-negative trees, suggesting a correlation of lipid metabolism with the resprouting, which can be an indication of the resiliency of these trees to OQDS. This is the first report concerning the resprouting of OQDS-infected olive trees in the Salento area.

## 1. Introduction

In the last decade, outbreaks of the *Xylella fastidiosa* subsp. *pauca* ST53 (XFP) bacterium, the causative agent of olive quick decline syndrome (OQDS), caused severe economic losses to the olive industry of Salento in Apulia, Italy [1]. OQDS causes leaf, twig, and branch die-back that leads to tree wilting. In the Salento region, more than 6,500,000 infected trees were considered dead [2]. XFP is efficiently spread within and among olive groves by the insect vector *Philaenus spumarius* (“Hemiptera”, “Aphrophoridae”), commonly known as the “meadow spittlebug” [3]. The infection reached orchards located both north and south of counties in which the disease was first observed (i.e., the Gallipoli area of Lecce province), with an estimated invasion front of about 10 km per year [4]. The trees in this region formed a continuum for many kilometers that promoted the effective spread of the infection [5]. Indeed, one XFP-positive olive tree can potentially transfer the bacterium to 19 additional trees in one year [6]. In addition, the local cultivars, Ogliarola Salentina and Cellina di Nardò, have increased sensitivity to XFP. The abundance of alternative host plants, including natural flora and ornamental shrubs, aids pathogen dispersal in the region [5,7]. As a result, XFP became endemic in this region of Italy a few years after it was first recorded [6,8]. In prior years, olive trees that did not completely wilt after pathogen attack frequently sprouted suckers at their base. While these trees usually survived, they subsequently succumbed to the infection as well [9,10,11]. In contrast, during the last years, some of the olive groves located in areas associated with the first XFP outbreaks in Salento (i.e., the Gallipoli area) had suckers that formed a crown. Field surveys allowed for the observation that the tree resprouting is currently occurring in many olive groves of different municipalities located in between and south of the Gallipoli and Otranto areas of Salento [12]. Such olive groves previously were severely infected during XFP outbreaks, with this causing extensive twig and branch diebacks. It is important to note that no curative and/or disease management strategy was performed in these groves [12]. Due to the resprouting, growers had trained such new vegetation to be productive. This phenomenon occurred in both young (30–35 years of age) and adult (>35 years of age) groves and often yielded flowers and fruit. A preliminary investigation found that XFP were still present in these trees [12,13]. Resprouting is a key response to stress and disturbance widely recognized in woody plant species [14,15]. Upon a stress, including biotic ones, plants can react by activating internal physiological mechanisms that allow for the maintenance of both the viability and plant functions [16]. The resprouting occurs in the meristematic tissues of apical, basal and below-ground buds [14]. Resprouting have been also recently found in oak trees affected by the “sudden oak death” disease caused by *Phytophthora ramorum* in California, and such a survival strategy has been retained as fundamental for the resilience of trees affected by this pathogen [17]. The resprouting phenomenon has also retained a crucial place in the global warming context since plant species capable of reacting to severe stress, such as a prolonged drought, show more possibilities to recover when compared with non-resprouting species [18]. Within this frame, the resprouting observed in olive groves that previously were severely attacked by XFP is a new phenomenon that deserves attention. A deeper understanding of the mechanism by which olive trees respond to XFP is required to identify the characteristics that distinguish XFP-negative, XFP-positive resistant, and XFP-positive susceptible resprouting olive trees from XFP-positive susceptible symptomatic ones. Lipids can act as infection and plant-defense determinants, mediating the crosstalk during the interaction [19]. Lipid content was recently assessed in XFP-positive resistant and XFP-positive susceptible olive cultivars (e.g., Leccino and Ogliarola Salentina) in the presence or absence of biofertilizer (Dentamet^®^) treatment. XFP-positive OQDS symptomatic plants modulate the production of free fatty acids (FFA) and oxylipins known as antibiofilm (e.g., linoleic acid, 18:2), bactericides (e.g., 13-HODE, 9,10-diHOME), and plant growth promoter and defense inducers (e.g., 9-KOD/TrE; linolenic acid, 18:3; 9-HODE) [20]. Specifically, 13-LOX products and jasmonic acid by-products (e.g., 13-HOTrE) appear to be associated with XFP susceptibility and quick decline. The satellite imagery and Normalized Difference Vegetation Index (NDVI) assesses a tree growth trend during a 5-year time course (2018–2022) in XFP-positive resprouting trees, wherein the current study investigated whether the levels of FFAs, oxylipins, and plant hormones [salicylic acid (SA) and jasmonic acid (JA)] in Ogliarola Salentina and Cellina di Nardò samples harvested at a single time point, i.e., 2022, correlated with their state of health: XFP-negative (H), XFP-positive symptomatic (I), or XFP-positive resprouting (R).

## 2. Results

### 2.1. Satellite Imagery and Normalized Difference Vegetation Index (NDVI) Analyses

The trend of similarity as obtained through satellite imagery and NDVI analyses is shown in Figure 1, which also presents the results of the Kruskal–Wallis test (*p* < 0.001) and identifies the most significantly different of state of health. It is evident that groups “R” and “H” shared similar index values, whereas “I” had significantly lower values. H indicates negative to XFP olive trees; I indicates positive to XFP and OQDS symptomatic olive trees; and R indicates positive to XFP and resprouting olive trees.

The NDVI values over the timeline from April 2018 to September 2022 exhibited distinct patterns across the three states of health (Appendix A and Figure 2). “I” shows the lowest NDVI values throughout the observed period, with a minimal upward trend, indicating poorer vegetation health and productivity. Appendix A shows a mean NDVI value ranging from 0.09 in 2018 to 0.09 ± 0.07 in 2022. The NDVI values for this state of health remained mostly below 0.10, except for occasional peaks slightly above this threshold. In contrast, “R” demonstrated a notable upward trend over the years. Initially, its NDVI values were comparable to those of “I”. However, from mid-2019 onwards, the NDVI values for this state of health started to increase steadily, surpassing those of “I” and approaching the values of the “H”. By 2021, the NDVI values for “R” aligned closely with those of “H”, indicating a significant improvement in vegetation health and productivity.

“H” exhibited the highest and most stable NDVI values throughout the timeline. The NDVI values for this state of health consistently leaned around 0.15 to 0.20, reflecting a healthy and productive vegetation state. The trendline was relatively flat, indicating stable vegetation health with minor fluctuations over the years.

The statistical significance of the differences in NDVI values among the state of health was confirmed by the Kruskal–Wallis test (*p* < 0.001). Post hoc analysis using Dunn’s test shows that the NDVI values for “I” were significantly different from those of both “H” and “R” (Figure 1). However, there was no significant difference between “H” and “R”, highlighting that while the latter showed more variability, its median NDVI values were comparable to those of “H”.

### 2.2. Xylella fastidiosa subsp. pauca PCR Quantification and Lipidomic Analyses

The qPCR quantification of XFP-positive symptomatic (I) and XFP-positive resprouting (R) is shown in Figure 3. XFP-positive symptomatic samples had a XFP load similar to resprouting ones. To determine whether prior results on lipidomic profile were associated with this newly identified phenomenon, the most expendable and “motile” lipid fraction, FFA and oxylipins, described as the “reactive” lipid species, was assessed [21], along with the defense-related hormones JA and SA. In Figure 4, it is possible to observe that while the cultivar (O and C) (*p* = 0.0076) and the cultivar x state of health (O/C × H/I/R) binary interaction (*p* = 0.0196) had a significant effect on the multivariate oxylipin and FFA profiles, the main effect of the state of health did not (*p* = 0.0796). It is indeed true that the main difference observed was between XFP-negative (H) and XFP- positive OQDS symptomatic plants (I), which were mostly separated along PC2. On the other hand, as also indicated by a significant interaction term, there was a different behavior of the two cultivars with respect to the I/R trees: while for Cellina di Nardò, the R trees, though being closer to the I trees, were still mostly distinguishable from them, this was not the case for Ogliarola Salentina, especially due to the high variability in the sampled I plants. The next step in ASCA is to determine the effect of individual design terms or their combination using principal component analysis (PCA). PCA was applied to the combination (X_health + X_(cult × health)) to determine how the state of health differentially impacted the lipid profiles of the two investigated cultivars. 

The scores and loadings plots are shown in Figure 4 and Figure 5. The scores plot revealed a clear separation between the H trees (of both cultivars) at positive values along PC2 and the I and R cultivars (at negative values along the same component) (Figure 4). The differentiation reported in Figure 5 was primarily driven by 13-HpOTrE, 13-HpODE, and 8-HpODE, which were higher in the H plants, and 20:4, DSF1 (16:1), 8,13-diHODE, SA, and 18:0, which were higher in the I and R plants. In the H plants, almost all metabolites with positive loadings along PC2 had a higher concentration in C than in O (Figure 4). The scores plot also revealed a greater difference between the I and R for cultivar O than I and R for cultivar C (Figure 4).

This was mostly because the I cultivar O trees had higher metabolite levels with loadings falling in the third quadrant of the plot (Figure 5). While achieving a similar result (resprouting), the lipid profiles of Cellina di Nardò and Ogliarola Salentina differed (Appendix A). This is not a novel finding, given that these two cultivars diverge both genetically and metabolically [22,23].

In Cellina di Nardò, 9- and 13-oxylipins were up-regulated in the XFP-negative trees, down-regulated in the XFP-positive trees (Appendix A), and presenting an intermediate phenotype in the resprouting ones (Appendix A). The defense-related hormones were down-regulated in XFP-negative as well as in XFP-positive trees, whereas they were up-regulated in the resprouting (Appendix A). In Ogliarola Salentina, the scenario changed. The 9- and 13-oxylipins (but mostly 9-) were up-regulated in the XFP-positive trees, were down-regulated in the resprouting trees, and presented an intermediate phenotype in XFP-negative trees (closer to resprouting ones). JA was up-regulated in the XFP-positive as well as in the resprouting trees, whereas it appeared down-regulated in XFP-negative trees (Appendix A). In Ogliarola Salentina, SA followed a pattern similar to Cellina di Nardò: down-regulated in XFP-positive as well as in XFP-negative trees and up-regulated in resprouting samples (Appendix A).

## 3. Discussion

Satellite imagery and NDVI analyses allowed us to ascertain a clear growing trend in the olive trees sampled for the study from 2018 to 2022 for H and R samples. It was noticed that the first outbreak of OQDS symptoms in the area appeared around 2009–2010 [12], which postulates that the initial resprouting started eight to nine years after the XFP outbreak. Satellite imagery and NDVI analyses confirmed their usefulness in assessing the vegetative status of olive trees in the Salento area upon the XFP outbreak [24]. Resprouting plants regenerate after severe loss of biomass by sprouting from meristematic tissues. Several studies report that the resprouting “is a species-specific characteristic (determined by the presence/absence of dormant meristematic tissue), whereas the expression of this trait is a function of the severity rather than the mechanism of damage”. Zeppel et al. [18] argue that plants resprout in response to any major disturbance, whether this is fire, herbivory, drought, or insect/pathogen attack [18]. To the best of our knowledge, there are no studies that link the resprouting with a specific metabolic pathway; in view of our previous results and with the evidence of this emerging re-sprouting phenomenon in OQDS-affected trees, we analyzed some free fatty acid and oxylipin in olive tree resprouting that was positive to the bacterial pathogen XFP with respect to XFP-negative or XFP-positive and OQDS symptomatic plants. 

The rapid flux of reactive lipid species into the cytoplasm is one of the earliest phenomena that occur during both pattern-triggered immunity (PTI) and effector-triggered immunity (ETI), associated with biotic or abiotic stress recognition. Changes in FFA, oxylipin, JA, and SA accumulation orchestrate downstream immune responses [25,26]. As we suggested elsewhere [27], FFA can be cleaved from more complex lipids [e.g., diacylglicerides (DAG) and phospholipids (PLs)] by lipases produced from the plant, in response to pathogen infection, as well as from *X. fastidiosa* [27,28]. In turn, these FFA can act per se (e.g., diffusible signal factors (DSFs) [27,29]) as modulators of quorum sensing (QS) or being further processed to oxylipins, which also act as regulators of QS (ODS oxylipin-dependent quorum sensing system) (e.g., 7,10 di-HOME) [29,30]. Plants use these oxylipins for diverse action [31]; the accumulation of 9- and 13-oxylipins modulate the plant response to OQDS [20,31]. In accordance with the results of qPCR, the I samples presented a similar amount of XFP with respect to R, suggesting that resprouting plants may not depend on reducing the pathogen load but may instead involve limiting the causes of extensive decline. According to targeted lipidomic profile, the oxylipins and FFA profile of Ogliarola Salentina and Cellina di Nardò was revealed as different in the H and I samples, and the two cultivars showed resprouting, combining the decrease in several oxylipins (e.g., 9-HODE) and increase in SA and JA. At least in Ogliarola Salentina, this trend is confirmed by 2020 lipidomic data [20]: 9- as well as 13-oxylipins were down-regulated in Xfp-negative trees compared to Xfp-positive ones. This lipid profile is suggestive of a less stressful (oxidant) habitat for *X. fastidiosa,* slowing the symptoms. It is the case that 9- as well as 13-oxylipins are correlated with defense hormone levels: notably both SA (indirectly) and JA (directly) levels are affected by 9- and 13-LOX pathways [32,33,34,35], demonstrating that SA-mediated pathways are activated when *X. fastidiosa* lacks its O-antigen or when the plant (i.e., grapevine) is primed with *X. fastidiosa* O-antigen, suggesting that the plant immune system can adapt to recognize a pathogen activating the SA pathway. This research team demonstrated that JA-mediated defense against wild-type *X. fastidiosa* occurs in local tissue early in the infection process and is maintained at a systemic level. The phenomenon observed in the resprouting samples should be associated with different aspects related to plant immune response, and it needs to be further investigated at the molecular level.

According to our results, we can hypothesize that the selective pressure may help trees in Salento area to favor pathogen recognition and the onset of a more efficient immune response. Troha and Ayres [36] found that “during co-evolution with pathogens, hosts acquired defensive health strategies that allow them to maintain their health or promote recovery (resilience) when challenged with infections”. These “cooperative defenses” can promote endurance or resilience by antagonizing or withstanding a pathogen, limiting physiological damage and promoting health without negatively impacting the pathogen’s ability to survive and replicate in its host. Understanding the cooperative defense mechanisms in plants will offer new perspectives on potential treatment options [36]. 

Satellite imagery and NDVI analyses allow us to suggest that olive tree XFP-positive re-sprouting has a positive trend, at least in the last 5 years. NDVI data confirm the trend. Regarding this, re-sprouting can be visioned as a larger resiliency phenomenon that is under way in the area, allowing plants to coexist with the pathogen. In the case that the resprouting phenomenon will last over time, we would need to identify and analyze the strengths, weaknesses, opportunities, and threats of the resprouting phenomenon, despite the fact that the resprouting plants can be a reservoir of XFP inoculum in the infected area but can nonetheless allow the preservation of the biodiversity by safeguarding the Ogliarola Salentina and Cellina di Nardò susceptible cultivars. 

## 4. Materials and Methods

### 4.1. Study Site and Sampling Procedures

From each OQDS symptomatic tree, mature shoots were collected in the cardinal points of the upper and middle parts of the canopy where branches affected by leaf wilting and twig dieback were observed. Samples consisted of fragments of 15–20 cm of two-year-old twigs (around 0.5 cm in diameter) not yet affected by the desiccation and taken from portions of the crown close to visibly wilted branches. The samples for the XFP-negative and XFP-positive and resprouting trees were collected similarly in the four cardinal points from the upper and middle part of the canopy. The study was performed with the following samples collected in April 2022: (i) 6 XFP-negative olive trees (named H) of Ogliarola Salentina (O) and 6 of Cellina di Nardò (C) (12 total H samples); (ii) 6 XFP-positive symptomatic olive trees (named I) of Ogliarola Salentina (O) and 6 of Cellina di Nardò (C) (total 12 I samples); (iii) 6 XFP-positive resprouting (named R) olive trees of Ogliarola Salentina (O) and 6 of Cellina di Nardò (C) (total 12 R samples). Each tree was approximately 50 years of age in the Gallipoli area (40°01′57.5″ N 18°03′58.9″ E), except the XFP-negative samples collected in the Taranto area (40°35′33.1″ N 17°24′12.01″ E). From this point, the samples are indicated as (1) H to indicate negative to XFP olive trees; (2) I to indicate positive to XFP and OQDS symptomatic olive trees; (3) R to indicate positive to XFP and resprouting olive trees; (4) C to indicate cultivar Cellina di Nardò; (5) O to indicate cultivar Ogliarola Salentina. In Appendix A, images are provided of the olive groves where the XFP-positive resprouts were collected (A), along with one of the olive trees of the Cellina di Nardò variety (B) and one of the Ogliarola Salentina variety (C) that were selected for sampling.

### 4.2. Satellite Imagery and NDVI Analyses

To investigate the primary productivity of the three state of health and discern any effects on the olive trees’ vigor, we used the Normalized Difference Vegetation Index (*NDVI*, Equation (1)) derived from remote sensing data.
(1)$$NDVI=\frac{\left(NIR-VIS\right)}{\left(NIR+VIS\right)}$$

The formula used for the calculation of the index. *NIR* and *VIS* are the reflectance measures acquired in the near-infrared and the red (visible) regions, respectively.

The images were obtained from the Copernicus Sentinel satellite, which provides images with a spatial resolution of 10 m (https://scihub.copernicus.eu/, accessed on 25 June 2024). Data were collected from 2018 to 2022, specifically during the seasonal period of vegetative growth of olive trees in the area (April to September). Images containing clouds, which impeded the index calculation, were excluded from the analysis. For each state of health, monthly NDVI values were averaged. All the computations and analysis of the satellite images were performed in Google Earth Engine [37].

Once all the NDVI rasters were calculated, a point grid was established on the study area, with each point corresponding to the centroid of a pixel. This allowed us to gather values for statistical analysis. To analyze the time series of NDVI, we performed the non-parametric Kruskal–Wallis test to detect any significant differences among the three series. This was followed by Dunn’s post hoc test to determine which state of health were similar or exhibited significant differences. The last statistical analysis was performed on R Software, version 4.3.3. [38].

### 4.3. DNA, Fatty Acid, and Oxylipins Analyses

For DNA extraction of H, I, and R plant samples, 1 g of fresh small pieces of petioles and basal leaf part were used. The DNA extraction was performed with a DNeasy^®^ Mericon™ Food Kit (Qiagen, Hilden, Germany) according to the PM 7/24 (5) *Xylella fastidiosa* [39]. Samples were assayed using real-time PCR by Harper et al. [40] according to PM 7/24 (5) *Xylella fastidiosa* [37] in technical duplicate to diagnose and quantify the bacterial pathogen (Figure 3). FFAs, oxylipins, and the plant hormones salicylic acid and jasmonic acid (SA and JA) were simultaneously extracted from 20 mg of lyophilized and grinded petioles and basal leaf following the method reported in Scala et al. [27]. Briefly, the solvent mix was added to the samples, constituting isopropyl alcohol/water/ethyl acetate (1:1:3 *v*/*v*), containing 0.0025% *w/v* of butylated hydroxytoluene to prevent peroxidation, as well as the internal standard 9-HODE_d4_ (Cayman Chemicals, Ann Arbor, MI, USA) at the final concentration of 5 µM (calculated on the final resuspension volume). The samples were mixed and centrifuged (12.000 rpm), the extraction procedure was repeated, and the clear supernatant was dried under nitrogen stream and resuspended with 100 µL of methanol. The extracts were analyzed by HPLC coupled to a Triple Quadrupole (6420 Agilent Technologies, Santa Clara, CA, USA) using Multiple Reaction Monitoring (MRM) for oxylipins and hormones, as well as Single Ion Monitoring (SIM) for FFAs, as reported in Scala et al. [27]. Extraction for mass spectrometry analysis was performed in triplicate for each plant sample, and each lipid extract was analyzed in three technical replicates. MRM data were processed using Mass Hunter Quantitative software (B.07.00 version, Agilent Technologies, Santa Clara, CA, USA). Appendix A report the normalized area of the FFAs (Appendix A), oxylipins, and hormones (Appendix A) analyzed in this study.

### 4.4. Statistical Analysis

ANOVA simultaneous component analysis (ASCA) was used to assess the impact of plant cultivar and state of health on FFA, oxylipin, SA, and JA levels. ASCA [41] is a multivariate generalization of ANOVA that can be applied to datasets with correlated descriptors in which, due to an ill-conditioned matrix, traditional multivariate analysis of variance (MANOVA) is not applicable. ASCA partitions overall variability in the mean-centered data matrix *X_c_* as the sum of the contributions of individual design terms (factors and interactions), plus residuals (*X_res_*) to account for variance not explained by the experimental design:(2)$$X_{c}=X_{cult}+X_{health}+X_{cult\times health}+X_{res}$$
where $X_{cult}$, $X_{health}$, and $X_{cult\times health}$ are the effect submatrices for the main effects of the cultivar and their state of health and binary interaction, respectively. Given the decomposition in (2), the multivariate effect of a design term is quantified by the sum of squares of the corresponding matrix elements. Its statistical significance is evaluated by comparing it with its null distribution, which is non-parametrically evaluated using a permutation test (in this study, this was carried out using 10,000 randomizations). The lipidomic profile of the same cultivar and the state of health were also plotted as a heatmap, in which the target compound/area of the internal standard 9-HODEd4 ratio is reported with Z-score normalization values, and hierarchical clusters are set for $X_{cult\times health}$.

## 5. Conclusions

An NDVI analysis from satellite imagery over the past five years in the OQDS-affected “red” zone of Salento has revealed that some olive trees positive for XFP and resprouting show NDVI values similar to those of XFP-negative plants. Interestingly, when examining the total amount of oxylipins produced by olive trees in three health states, the resprouting samples (R) exhibited a significantly lower amount of oxylipins compared to the infected (I) ones (Appendix A). To the best of our knowledge, no studies have yet associated NDVI analysis with lipid analysis in the field. NDVI is based on an algorithm that considers NIR and RED-light reflectance values. Recently, NIR has been used at a proximal level to detect lipid oxidation traits in plants and vegetable oils [42,43]. It can be suggested that the observed differences in NDVI, which distinguish R from I olive trees, might be partly influenced by lipid oxidation. If confirmed, this study could pave the way for further exploitation of satellite imagery in combination with metabolic analysis to monitor plant health.

## Figures and Tables

**Figure 1 plants-13-02186-f001:**
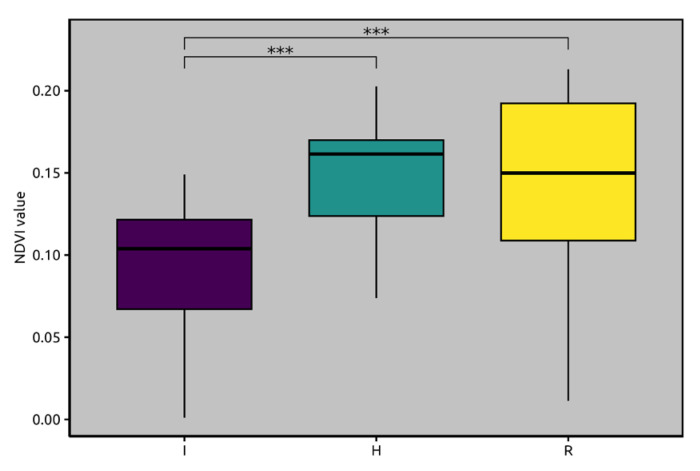
Box plots summarizing the mean NDVI values obtained through satellite imagery analyses for each state of health, highlighting the results of the statistical tests (Kruskal–Wallis test and Dunn’s post hoc test) and the significance level among the groups. H indicates negative to XFP olive trees; I indicates positive to XFP and OQDS symptomatic olive trees; R indicates positive to XFP and resprouting olive trees. ***: statistically significant.

**Figure 2 plants-13-02186-f002:**
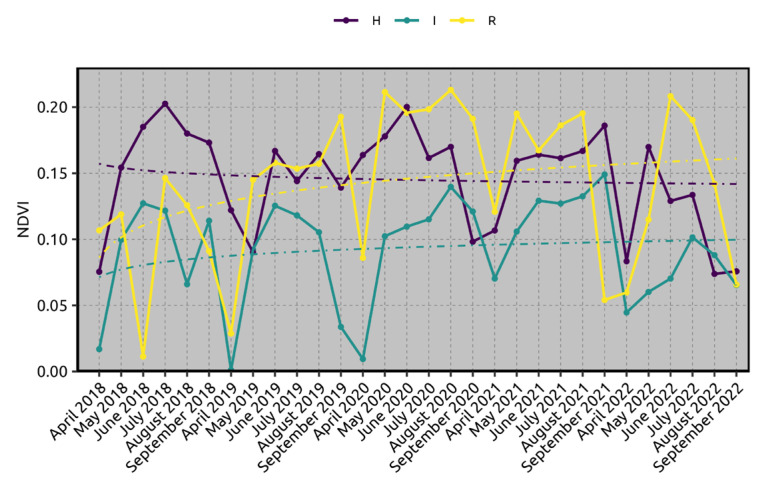
NDVI analysis chart as obtained through satellite imagery for 2018–2022. For each condition of state of health, an average logarithmic trendline is shown. H indicates negative to XFP olive trees; I indicates positive to XFP and OQDS symptomatic olive trees; R indicates positive to XFP and resprouting olive trees.

**Figure 3 plants-13-02186-f003:**
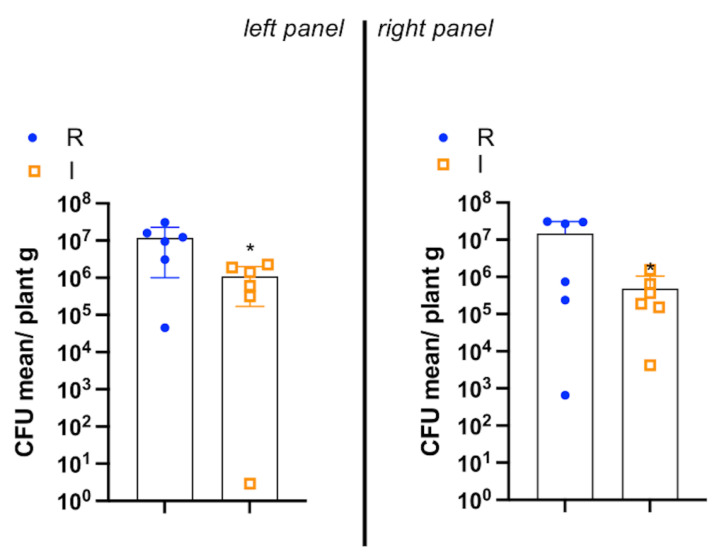
Box plot representation of XFP qPCR quantification of I and R. In the left panel of the figure are the results of 6 trees of the cultivar Cellina di Nardò; in the right panel of figure are the results of 6 trees of the cultivar Ogliarola Salentina. Data are expressed as CFU mean ± S.D. (* *p* < 0.02, R vs. I). I indicates positive to XFP and symptomatic olive trees; R indicates positive to XFP and resprouting olive trees.

**Figure 4 plants-13-02186-f004:**
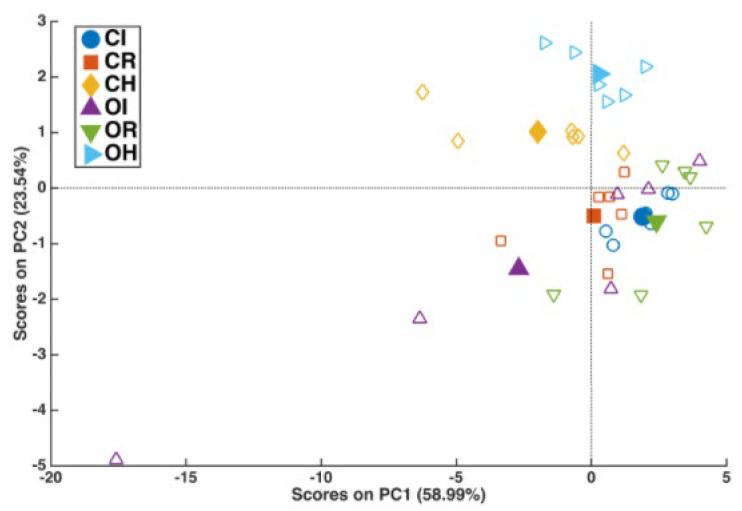
ASCA analysis. Scores plot of the PCA on the matrix resulting from the sum of the contribution of the state of health (H, I, R) and its interaction with the cultivar (C and O). The filled symbols in the plot represent the average scores of each group of trees, whereas the empty symbols account for the individual variability within each group (scores of individual trees). H indicates negative to XFP olive trees; I indicates positive to XFP and OQDS symptomatic olive trees; R indicates positive to XFP and resprouting olive trees; C indicates Cellina di Nardò; O indicates Ogliarola Salentina.

**Figure 5 plants-13-02186-f005:**
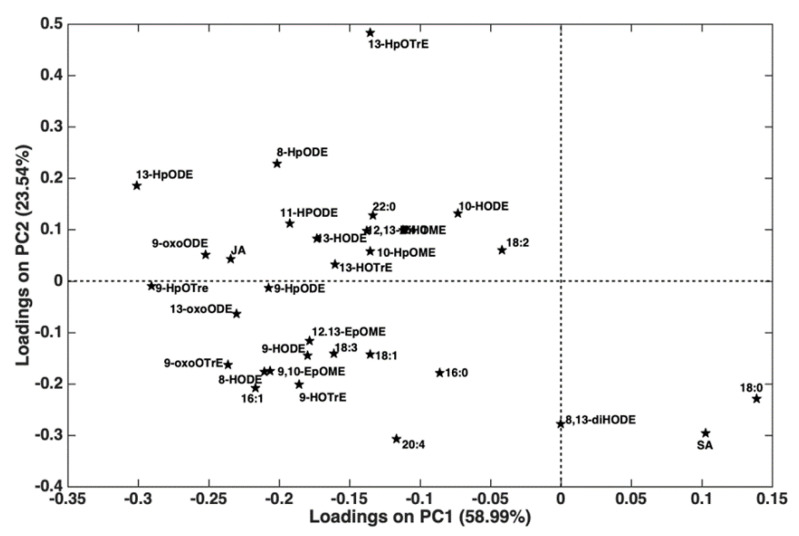
ASCA analysis. Loading plot of the PCA on the matrix resulting from the sum of the contribution of the state of health and its interaction with the cultivar. H plants were characterized, on average, by higher amounts of the metabolites with positive loadings on PC2, while R/I plants showed higher concentrations of the lipids with negative loadings on the same component. Palmitic acid (C16:0), palmitoleic acid (C16:1), stearic acid (C18:0), oleic acid (C18:1), linoleic acid (C18:2), linolenic acid (C18:3), arachidonic acid (C20:4), behenic acid (22:0), lignoceric acid (C24:0), hydroxyoctadecenoic acid (HODE), hydroperoxyoctamonoenoic acid (HpOME), hydroperoxyoctadienoic acid (HpODE), dihydroxyoctamonoenoic acid (DiHOME), epoxyoctamonoenoic acid (epOME), hydroxyoctatrienoic acid (HOTrE), hydroperoxyoctatrienoic acid (HpOTrE), oxo-octadecenoic acid, (OxoODE), and oxo-octadecatrienoic acid (OxoOTrE). Notation of the FAs and oxylipins (OM/D/TrE) is reported as the carbon number (CN) and the number of double bond (DB) equivalents.

## Data Availability

Data are contained within the article and Appendix A.

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
