# Peer review of "Assessment of Fatty Acid and Oxylipin Profile of Resprouting Olive Trees Positive to Xylella fastidiosa subsp. pauca in Salento (Apulia, Italy)"

_plants, 2024, doi:10.3390/plants13162186_

Round 1

Reviewer 1 Report

Comments and Suggestions for Authors

The authors in the present study entitled " Assessment of fatty acid and oxylipin profile of resprouting olive trees positive to Xylella fastidiosa subsp. pauca in Salento (Apulia, Italy)" address to a significant improvement in vegetation health and productivity from 2018 to 2022 of trees infected by Xylella fastidiosa subsp. pauca in 2013 outbreak when were anlyzed by satellite imagery and Normalized Difference Vegetation Index. Multivariate analysis revealed that lipid markers of resistance (e.g., 13-HpOTrE), along with jasmonic and sali cylic acid, were accumulated differently in the XFP-positive resprouting trees respect to XFP-positive OQDS symptomatic and XFP-negative trees, suggesting a correlation of lipid metabolism with the resprouting which can be an indication of the resiliency of these trees to OQDS.

In relation to the methodology, the material and methods content is clear and well redacted. The methodology has been properly applied to approach the proposed objectives. I have non-specific comments on the methodology.

About the results, material presented is clear and helps in their comprehension. Specific comments of the results:

- In figure 2, change the average logarithmic trendline with another trace, to differentiate from NDVI values.

- Some symbols in Figure 4 are not well recognized. Fix it.

The discussion is well-argued, focusing on these findings and contrasting them with relevant and current literature. The novelty of this paper lies in the observed correlation between the expression of lipid markers of resistance and the accumulation of jasmonic and salicylic acids, with the improve of resprouting of infected trees. These markers were found to be accumulated differently in XFP-positive resprouting trees, suggesting their participation in the resistance process. Understanding the cooperative defence mechanisms in plants will offer new perspectives on potential treatment options or also in improve breeding programs to get resistence selections.

Author Response

In attach our answers.

Reviewer 2 Report

Comments and Suggestions for Authors

The present document evaluates the composition of olives trees resprouting after the contamination with Xylella fastidiosa.

The research is original mixing data from satellite to follow the resprouting progress trees with biochemical analysis.

The introduction provides a good vision of the litterature already done on the topic and presents quite well the research question, besides it's doubled, it would be interesting to better highlight the interest of following the trees by satellite combined with the chemical analysis as it's not completely explicit. The referenced used are of good quality.

The results are well presented and treated, only the figures 1&2 legends should be modified. One thing that I do not understand why you call "site" the regrowth modalities, that are the analysis of the two varieties, several trees by variety and mainly one location. I recommend to change this term both in the figures and in the text.

The discussion is complete with good references.

The material and method section is globally complete, except when it comes to the analysis of FFA, Oxylipins and hormons. You do not indicate in which type of vegetal material you conducted the extraction and how the extraction was made for each metabolite. It should be completed.

More details on the attached document

Author Response

In attach our answers.

Round 2

Reviewer 2 Report

Comments and Suggestions for Authors

The authors have now provided a corrected version of the document, they have notably completed the material and method section as requested which was the main problem of the preceden version. They have modified the denomination of the modalities studied. And they have include conclusions which open the topic.